# Combination of Two Photosensitisers in Anticancer, Antimicrobial and Upconversion Photodynamic Therapy

**DOI:** 10.3390/ph16040613

**Published:** 2023-04-19

**Authors:** Martina Mušković, Rafaela Pokrajac, Nela Malatesti

**Affiliations:** Department of Biotechnology, University of Rijeka, Radmile Matejčić 2, 51000 Rijeka, Croatia; martina.muskovic@biotech.uniri.hr (M.M.); rafaelapokrajac1@gmail.com (R.P.)

**Keywords:** photosensitiser, anticancer photodynamic treatment, antimicrobial photodynamic treatment, upconversion nanoparticles

## Abstract

Photodynamic therapy (PDT) is a special form of phototherapy in which oxygen is needed, in addition to light and a drug called a photosensitiser (PS), to create cytotoxic species that can destroy cancer cells and various pathogens. PDT is often used in combination with other antitumor and antimicrobial therapies to sensitise cells to other agents, minimise the risk of resistance and improve overall outcomes. Furthermore, the aim of combining two photosensitising agents in PDT is to overcome the shortcomings of the monotherapeutic approach and the limitations of individual agents, as well as to achieve synergistic or additive effects, which allows the administration of PSs in lower concentrations, consequently reducing dark toxicity and preventing skin photosensitivity. The most common strategies in anticancer PDT use two PSs to combine the targeting of different organelles and cell-death mechanisms and, in addition to cancer cells, simultaneously target tumour vasculature and induce immune responses. The use of PDT with upconversion nanoparticles is a promising approach to the treatment of deep tissues and the goal of using two PSs is to improve drug loading and singlet oxygen production. In antimicrobial PDT, two PSs are often combined to generate various reactive oxygen species through both Type I and Type II processes.

## 1. Introduction

Recent decades have been marked by the evolution of photodynamic therapy (PDT) into a viable treatment for a variety of solid tumours. PDT is a minimally invasive and clinically authorised therapeutic strategy that selectively applies cytotoxicity against cancer cells or other abnormal tissues [1,2,3]. Ancient civilisations used light for therapeutic purposes in medicine, but the idea of phototherapy was mostly forgotten for several centuries before re-emerging in the early twentieth century, through the work of Niels Filsen, Oskar Raab and Herman von Tappeiner [4]. Niels Finsen conceived of the term “phototherapy”, meaning the use of light for medicinal purposes, after finding that exposure to red light inhibits the development of smallpox pustules and that it could be a promising treatment option for the disease. Moreover, his use of ultraviolet (UV) light for the treatment of cutaneous tuberculosis marked the beginning of contemporary phototherapy; Finsen received the Nobel Prize for his discovery in 1903 [5]. In addition to Finsen’s revelations, researchers discovered that cell death can be induced by the combination of light and certain chemical substances. It was first reported in 1900, by Oskar Raab, that light combined with acridine dye has toxic effects on paramecia. Raab noticed that the cytotoxic effect of acridine was more efficient in sunlit conditions and was minimal on days on which there were thunderstorms. These observations led him to the conclusion that exposure to light activates acridine dye and potentiates its cytotoxic activity against paramecia [6]. Raab’s serendipitous discovery was later adopted in the form of the general approach of PDT—the use of dye as a photosensitiser (PS). In 1903, Herman Von Tappeiner and Albert Jesionek described the treatment of skin tumours with the topical application of dyes, such as eosin, in combination with white light. They subsequently published clinical data on the application of PS as a treatment for skin cancer, lupus, and other pathological incidences in 1905, describing the phenomenon as “photodynamic action” [7]. In 1904, Tappeiner and Albert Jodlbauer noted that oxygen is the key element in successful photosensitisation, which was then described by Tappeiner using the term “photodynamic therapy” [8].

### 1.1. PDT—Mechanism of Action

The standard PDT process begins with the introduction of a PS, which is expected to accumulate in malignant tissue, followed by irradiation with light of a specific wavelength that is correlated with the absorption spectrum characteristics of the administered PS [9]. A key element in PDT, however, is the molecular oxygen in its ground state (^3^O_2_) present in the malignant tissue. The interaction of ^3^O_2_ with PS activated by light generates singlet oxygen (^1^O_2_), a highly reactive, excited state of molecular oxygen [10]. The PDT is initiated when the PS in the target tissue absorbs light, which sets off a sequence of photophysical and photochemical processes that result in the production of reactive oxygen species (ROS) (Figure 1) [11]. Following light absorption, the PS is converted from its stable, ground state (^1^PS) to a singlet state that is electronically excited (^1^PS*) and has a brief lifetime (a few nanoseconds) [3]. The highly unsTable 1PS* may decay into the ground state in two ways: by losing surplus energy via photon emission as light energy (fluorescence), or by emitting energy as heat (vibrational relaxation) [11]. Alternatively, the ^1^PS* can convert to the triplet excited state (^3^PS*), in a process known as intersystem crossing (ISC), through a spin change in the electron in the higher-energy orbital. The PS in the excited triplet state, unlike its singlet counterpart, has lower energy and a longer lifetime (tens of milliseconds), providing the PS with sufficient time to directly exchange energy with its surroundings [12]. The decay of the ^3^PS* into the ground state can occur through photon emission (phosphorescence) or through Type I and Type II reactions, both of which require interaction with the molecules in the immediate vicinity [3,5]. The Type I process occurs if the ^3^PS* reacts with biomolecules in its vicinity, such as lipids, the amino acids in proteins, and nucleic acids, forming the superoxide radical anion (O_2_^•−^) through electron transfer. This leads to the formation of hydrogen peroxide (H_2_O_2_), a precursor of hydroxyl radical OH^•^, which is highly reactive and capable of interacting with nearly all biomolecules [13]. Soon after this, a Type I reaction yields ROS through several cascade reactions between the excited PS, oxygen, and biological substrates present in the cell; this may be a promising route for achieving better outcomes under hypoxic conditions [14]. In Type II reactions, the ^3^PS* transfers its energy to ^3^O_2_, leading to the formation of ^1^O_2_ [15]. In this highly reactive state, ^1^O_2_ can target many biomolecules, which results in oxidative damage (Figure 1) [12,15]. The Type II process is the predominant mechanism of oxygen-dependent PDT, although both reactions may take place simultaneously, depending on the PS in use and the concentrations of certain substrates and molecular oxygen in situ [15].

### 1.2. Cell Death Pathways and Intracellular Localisation in Anticancer PDT

In 1991, Oleinick’s group published the first report that described an apoptotic cell death pathway related to PDT. This group observed a rapid initiation of apoptotic cell death after the illumination of cells containing PS through a process involving characteristic apoptotic events such as DNA cleavage and subsequent cell fragmentation [16]. Later, the apoptosis caused by a release of cytochrome *c* from the mitochondria to the cytoplasm was described, as well as the apoptotic outcome triggered by direct damage to Bcl-2, a protein that regulates cell proliferation [17,18]. There are several other mechanisms triggered by PDT, particularly when the PS is localised in the mitochondria, that ultimately lead to apoptotic cell death [16,19]. A necrotic cell death pathway, characterised by swelling of the cell membranes, chromatin condensation and an erratic DNA degradation pattern, is often associated with high-dose PDT. Such an approach is generally avoided since it hinders selectivity and could cause damage to surrounding healthy tissue [19]. On some occasions, the PDT-induced damage can lead to autophagy, a type of cell degradation which includes the degradation and recycling of photodamaged organelles before they can trigger apoptosis. It usually occurs at a low PDT dose and may have a cytoprotective effect against photodamage [20]. When PDT targeting involves the endoplasmic reticulum (ER), the photodamage will likely initiate paraptosis. This outcome can be distinguished by the occurrence of cytoplasmic vacuolation, as well as by the absence of usual apoptotic morphology [20]. Paraptosis was originally identified as a response to photodamage in low-dose PDT protocols, which is possibly associated with the MAP kinases pathway, and it could be an alternative cell death pathway for malignant cells that are not responsive to PDT due to defective apoptosis [21].

Effective PDT is greatly dependent on the uptake of PS by targeted cells. Since ^1^O_2_ and other ROS have a short half-life, cellular localisation of the PS influences what type of photodamage will occur after illumination. A variety of phototherapeutic agents with considerably different structures have had their intracellular distributions identified. Among the relevant structural characteristics of the PS are its charge (whether is anionic, neutral or cationic), lipophilicity (expressed as log*P*, octanol/water partition coefficient) and the degree of asymmetry in the molecule [15]. For example, a hydrophobic anionic PS with a small number of charges may have high cellular uptake due to its diffusion through the plasma membrane and subsequent relocation to the intracellular membranes, whereas those with more than two negative charges are often too polar to enter the cell via diffusion and therefore must be taken in by endocytosis [15]. 

The intracellular distribution of the PS is now much easier to determine due to confocal laser scanning fluorescence microscopy, a method used to acquire high-resolution optical images by capturing several two-dimensional images in different depths of the sample, which are afterwards reconstructed into three-dimensional structures [15]. The localisation site can therefore be pinpointed by using organelle-specific probes whose fluorescence is distinct from the PS [22]. Moreover, such probes can be applied to distinguish damaged parts post-irradiation. The initial and post-irradiation localisation of the PS are important to monitor since different photosensitising agents are going to inflict photodamage at different intracellular locations, and their preferred site of accumulation can direct the outcome of cell death towards apoptotic or necrotic responses [23]. Aside from confocal laser scanning microscopy, other techniques such as fluorescence resonance energy transfer (FRET) can also be employed to establish the intracellular placement of the PS [24].

Mitochondria are the relevant organelle targets in the cell for a PS applied in PDT [25], and many PSs promote apoptosis by inciting mitochondrial damage upon irradiation, e.g., benzoporphyrin derivative (BPD-MA) [26]. Photosensitisers, as well as their formulations with cationic charges and hydrophobic properties, are likely to be preferentially localised in mitochondria, thanks to electrostatic interactions and an affinity for the lipid environment [15,26]. Lysosomal photodamage can lead to loss of cytochrome *c*, which triggers apoptosis, by inducing several cascade reactions in the cell [27,28]. Lysosomal localisation was observed for PSs with anionic charges [29]. Although PSs with lysosomal targeting are often found to be less effective than others, they can still initiate a significant amount of photodamage [30]. Plasma membrane localisation is a relatively uncommon PDT target, and PS accumulation often occurs briefly after illumination, after which the PS can distribute inside the cell and finally accumulate in the Golgi complex, as shown, for example, with Photofrin^®^ [31]. Two typical signs of apoptosis, DNA fragmentation and phosphatidylserine externalisation, were not noticed in this specific PDT protocol [31].

A recent review from 2022 describes the characteristics of the non-traditional types of cell death in anticancer PDT (namely, paraptosis, pyroptosis, parthanatos, necroptosis, ferroptosis and mitotic catastrophe) compared to apoptosis, necrosis and autophagy [32]. The aim of the review was to identify the most successful pathway, and it was concluded that immunogenic cell death (ICD) is the most promising one that can be achieved by mixing cell death mechanisms and by combining high and low PDT doses and different PSs [32].

### 1.3. Vascular Damage and Immune Response in PDT

Aside from its cytotoxic effects that result in localised cell death, PDT can effectively target tumour vasculature as well. When the PS accumulates in the endothelial cells of tumour blood vessels, cellular damage can occur because of ROS generation [3]. A disrupted vasculature and its obstruction by blood clots can inhibit or significantly reduce the flow of nutrients to tumour cells [33]. Several in vivo studies have shown that damage to microvasculature after PDT resulted in severe tissue hypoxia and anoxia [34,35]. A preclinical study involving pyropheophorbide-*a* showed that a vascular response occurs in two phases: the first is vasoconstriction, which can be noticed immediately, and the second is characterised by thrombus formation [36]. A similar effect was also observed with some other PSs, such as BPD-MA and Photofrin [34,35,37]. Vascular PDT has the advantage of using PSs that have rapid systemic clearance and cause minimal skin hypersensitivity while achieving high long-term efficacy in tumour eradication [33,38]. 

It is known that traditional approaches to treating cancer, such as chemotherapy, radiotherapy and surgical removal of tumorous tissue, determine a patient’s immunosuppressive response. On the contrary, PDT is likely to generate acute inflammation and promote the discharge of pro-inflammatory cytokines, as well as the attraction of neutrophils, mast cells and macrophages at the inflammation site [33,39]. Moreover, tumour-derived antigens should consequently be presented to T-cells, which in turn activate anti-tumour adaptive immunity [39]. When PDT initiates necrosis, constituents of cytosol spill into the extracellular area and trigger a potent immune response. Immunity could thereafter be potentiated by attracting leukocytes to photodamaged tumour sites and escalating tumour-generated antigen presentation [40]. Certain PSs, such as Photofrin, lead to a noteworthy increase in interleukin-6 (IL-6) expression, a cytokine relevant for complement activation, the release of acute-phase proteins and neutrophil migration [41]. In addition, IL-16 is involved in the promotion of an adaptive immune response [42]. 

Immunogenic cell death (ICD) is a cell death modality that provokes both adaptive and immune responses, and it can stimulate an anticancer immune response that can increase the efficiency of PDT and activate anticancer immunity [43]. The production of ROS and the general stress caused by photodamage followed by the exposure of damage-associated molecular patterns (DAMPs) is a prerequisite for ICD, and an important function of DAMPs in vivo is the stimulation of dendritic cells (DCs), antigen-presenting cells that promote adaptive immunity [44]. The maturation of DCs stimulates CD 4+ CD8+ and T-cells proliferation, followed by the production of cytotoxic cytokines such as the tumour necrosis factor (TNF-α) and interferon γ (IFN-γ), which develops an adaptive immune response [43]. However, given that tumours do not possess plenty of DCs, some therapeutic strategies involve the injection of DCs directly into them. Dendritic cells containing tumour-derived peptides, genes, proteins or other components, have been examined as potential anti-cancer vaccines, and the activation of DCs by PDT-treated cell material may improve treatment outcomes. For example, a combined approach with 5-aminolevulinic acid and DCs injected into PDT-treated tumours had been shown to obtain improved survival compared to either method alone [45].

### 1.4. Light Sources in PDT and Types of Photosensitisers

Lasers and laser-emitting diodes (LEDs), as well as incandescent light sources, have so far been used to carry out PDT. Laser light sources are typically expensive and necessitate the use of an additional optical system when a larger region needs to be irradiated [46]. Other light sources can be utilised in conjunction with optical fibres, with the appropriate wavelength set, to reach the target tissue for irradiation, but their disadvantage may be the heat effect, which must be avoided in PDT [46,47]. When discussing the penetration of light into the tumour, it must be noted that the complex nature of malignant tissue causes the light to be reflected, scattered, and absorbed, all of which depends upon the properties of tumour tissue as well as the wavelength in use [48]. The chromophores that are present in tissues (haemoglobin, myoglobin, collagen, melanin, etc.) absorb light of lower wavelengths, thereby competing with the PS and possibly reducing the efficiency of PDT. The penetration of light through tissue increases with increasing wavelength, so the optimal wavelength range, 600–1200 nm, is often recognised as the tissue “optical window” [11]. In PDT, shorter wavelengths (<600 nm) cause increased skin photosensitivity due to high absorption and lesser tissue penetration, whereas wavelengths above 850 nm lack sufficient energy to excite ground state oxygen to its singlet state. For that reason, the optimal tissue permeability for therapeutic purposes spans from 600 to 850 nm, often referred to as the “phototherapeutic window” [49].

In addition to choosing the appropriate wavelength and light supply, a successful therapeutic outcome requires a clearly defined light dosimetry, which consists of light fluence and its rate, and a set duration of illumination. Light fluence (expressed in J/cm^2^) is defined as the total energy of applied light on a defined area, whereas the light fluence rate (expressed in mW/cm^2^) is the incident energy per time unit over a defined area [50,51]. It has been reported by several studies that lower light fluence rates are favourable for PDT, particularly when the apoptosis of cancer cells is the preferred outcome [50,51,52,53,54]. Higher light fluence rates, on the other hand, are experimentally proven to be related to cell death pathways that mostly result in necrosis and, consequently, local inflammation and swelling (oedema) [46,47,51,54]. When calculating the PDT dosimetry, two main methodologies, implicit and explicit dosimetry, are usually applied. Explicit dosimetry includes the measurement of the individual components that are involved in the treatment (e.g., the concentration of the PS, ground state oxygen levels, delivered light dose), while implicit dosimetry relates to the measurement of the photobleaching of the PS in the system, which involves all treatment response factors. Furthermore, the biophysical and biological tissue responses, such as the vascular shutdown and tumour necrosis that are induced by the treatment, and singlet oxygen luminescence dosimetry, known as direct dosimetry, are also included in clinical PDT dosimetry [55,56,57].

In antimicrobial PDT, the main classes of photosensitisers used so far are phenothiazinium derivatives, porphyrins, chlorins, phthalocyanines, xanthenes, fullerenes, as well as riboflavin, curcumin and phenalone derivatives [58]. Since antimicrobial PDT includes a wide array of PSs spanning several categories of organic compounds, various wavelengths of light can be employed as well. For example, phenothiazinium derivatives, such as methylene blue, have strong absorption of red light, making them favourable for medical use due to higher tissue penetration of longer wavelengths [58]. Certain PSs are activated by irradiation with the UV/blue part of the spectrum (<480 nm), such as phenalones, while xanthenes have a higher absorption in the green part of the spectrum (480–550 nm) [59,60]. While these PSs may not be ideal for use in tissues, they may be a promising choice for disinfection purposes [61]. 

Nearly all PSs for anticancer PDT are cyclic tetrapyrroles, and based on their photophysical and photochemical properties, the main groups are porphyrin derivatives, chlorins, phthalocyanines and porphycenes [62]. In the literature, PSs are often classified based on when they were created. They are separated into three generations, each one standing for a relevant novel approach that is significantly different than the earlier one. The first-generation PSs includes hematoporphyrin derivative (HpD) and its analogues, while the second generation of PSs is a structurally diverse category that includes porphyrins, modified porphyrins, chlorophyll derivatives and dyes. Third-generation PSs are compounds from the first and second generations modified to achieve higher tumour tissue targeting using methods such as antibody conjugation or assembling into nanoparticles [63,64]. It is important to note, however, that such classification does not unequivocally imply that the most recently developed PSs are the ones with the best clinical properties; furthermore, most of the PSs of the second and third generation are not yet clinically available [63].

Photosensitisers can be differentiated in a clinical sense by their targeting, and such an approach to classification can be considered on two levels. Firstly, when considering its tissue accumulation, a PS can target either neoplastic tissue or nearby neovasculature. Secondly, photosensitisers can be classified by their intracellular targeting, as they can specifically accumulate in the cell membrane, subcellular membranes and other parts of the intracellular structure [63]. Third-generation modified PSs can also be modified in such a way to increase intracellular targeting specificity [65,66]. 

### 1.5. Limitations of PDT with One PS and Combinations with Other Therapies

In recent years, PDT has been growing in popularity due to its many beneficial characteristics and possible multifunctionality, among which the most important are targeting tumour tissue and its vasculature, provoking an immune response and a low susceptibility to the resistance mechanisms of the tumour cells. However, despite its high selectivity and lower toxicity in comparison to traditional oncologic approaches, such as chemotherapy and radiotherapy, as well as better overall post-treatment cosmetic results, PDT still has plenty of space for development. Even those PSs with relatively rapid clearance rates tend to stay in the system for a significant amount of time and cause photosensitivity, which requires avoiding natural and, in some cases, even artificial light [3]. Furthermore, some PSs may have promising properties in vitro considering their singlet oxygen yield, but the hypoxic environment of cancerous tissue hinders the generation of ^1^O_2_ [3]. Due to the relatively poor selectivity for cancer cells of first-generation PSs, novel approaches to targeting and accumulation in the tumour tissue are constantly being developed. There is also a significant interest in gaining control of action, which must remain in a very narrow tissue area to avoid unnecessary damage to the surrounding, healthy tissue. Hence, further efforts in setting up proper dosimetry protocols for PSs in clinical use are of great importance [62]. Lastly, there is a growing interest in antimicrobial PDT, which has been used to eradicate pathogens in vitro, but its application to treat infections in patients or animal models has not yet seen considerable development [67]. 

Given that there is still no ideal PS, PDT with the use of only one PS as a single anticancer therapy is likely to have varying limitations to some degree, among which the most notable so far are those associated with oxygen dependence and hypoxia, light penetration depth and tumour size (lesion area) and developing resistance [68]. It is increasingly being confirmed that for effective anticancer outcome and long-term cure, PDT needs to target cancer cells (cellular targeting) [69], tumour vasculature (vascular targeting) [70] and immune responses [71], in addition to targeting other components of the tumour microenvironment (TME) [72]. This can be achieved by combining different agents and protocols. There are already many examples of combinations of two, or even more, different therapies, in which one is PDT. The use of other approaches in combination with PDT will achieve a combined effect that will lead to greater therapeutic success. Without going into details, since there are already many reviews, PDT can be combined with chemotherapy [73,74,75,76], radiotherapy [77,78,79], surgery [80,81,82,83], conventional immunotherapy [84,85,86], photothermal therapy (PTT) [87,88,89] and sonodynamic therapy (SDT) as sono-photodynamic therapy (sono-PDT) [90,91]; PDT can also be combined with more than one other therapy and diagnostic possibilities, which is especially being researched through nanotechnology [92,93,94]. The newly developed, and particularly promising, near-infrared photo-immunotherapy (NIR-PIT), which consists of near-infrared (NIR) dye conjugated with monoclonal antibodies, is progressing rapidly through phases of clinical trials, has shown efficacy on metastases and can be combined with other therapies [95,96,97,98]. In antimicrobial treatments, there are also examples of combining PDT with other therapies and agents such as antibiotic [99], antiviral [100] and antifungal agents [101]. The main goal of using PDT in these combinations is to overcome the limitations of single therapies, minimise side effects and achieve a greater overall therapeutic effect. 

A special approach in the combination of PDT against cancer with other therapies, called photodynamic priming (PDP), is the use of sub-lethal PDT doses to sensitise the entire TME, which consists of tumour cells, stroma and the vascular system, as the main treatment modality [102]. Since PDP can affect and prime distinct parts of the TME, not just the targeted tumour cells, it can improve tumour permeability to increase drug uptake, which is especially promising in combination with immunotherapy [102]. 

Another specific approach in achieving the combined effect in the treatment is to induce different mechanisms of cell death. So far, PDT has been linked to many different mechanisms of cell death, most often to apoptosis, necrosis and autophagy, but also the aforementioned, paraptosis, ferroptosis, pyroptosis and so on. It is known that, for example, PDT dose levels and the PS’s cell localisation and targeting of individual organelles may affect the type of cell death [102,103,104,105,106]. Different organelles react differently to ^1^O_2_ and other types of ROS. However, there is no agreement so far on which organelle would be the best to target in PDT, as it seems that targeting mitochondria, endoplasmic reticulum (ER), lysosomes or multiple organelles at the same time are all effective; it also must be taken into an account that some PSs change their localisation after photoactivation [15,102]. Targeting mitochondria and ER in PDT most often leads to apoptosis, although, in some cases, there is also evidence of paraptosis, while lysosomes are much more difficult to associate with a particular type of cell death. However, their targeting in sequence and prior to mitochondrial targeting/PDT can potentiate the loss of mitochondrial membrane potential and apoptosis, which may be useful for combined approaches [102,107]. These insights have certainly contributed to the development of the concept of combining two PSs in PDT, which will be discussed in more detail in the following sections.

In PDT with one PS (from now on referred to in the text as “single-PS PDT”), the combination of different tumour targets between cellular, vascular and immune responses can be achieved by changing the light parameters, i.e., fluence rate, duration of irradiation giving total fluence (“light dose”) and time point of irradiation after incubation of the PS, which is known as a drug–light interval (DLI) [55,56]. For example, the combination of a short and long DLI can be used to target both the tumour vasculature and the cells to enhance the PDT’s effect. In one such study, verteporfin (benzoporphyrin derivative monoacid ring A, BPD-MA) was injected twice into an R3327-MatLyLu rat with a prostate tumour, and the animal was irradiated with light after 15 min (short DLI) or after 3 h (long DLI) of incubation [108]. Different sequences of PS and light delivery were also investigated and, among the combination treatments that were both efficient and safe for normal tissue, the one with a long DLI followed by a short DLI had the most enhanced PDT efficiency [108]. In another approach, a combination of low and high light doses was used with either 2-[1-hexyloxyethyl]-2-devinyl pyropheophorbide-*a* (HPPH) or Photofrin to elicit an immune response, in addition to the direct attack on the tumour by ROS [109]. In all the experiments, 0.4 μmol/kg of HPPH or 5 mg/kg of Porphyrin were injected into BALB/c mice with implanted murine colon (Colon26-HA) and mammary (4T1) tumours, and the PS was incubated for 18–24 h, followed by irradiation at 665 nm (HPPH) or 630 nm (Photofrin) at a 14 mW/cm^2^ fluence rate [109]. A low light dose (48 J/cm^2^) was shown to induce an immune response more efficiently than a high dose (132 J/cm^2^), which was more efficient in a tumour growth control. The enhancement of long-term tumour growth control and persistent anti-tumour immunity was achieved with a combination where low light dose PDT was applied before high dose PDT, and this regime was successful with both PSs and on both tumour types [109].

#### Nanocarriers for Combinations of PDT with Other Therapies

Nanocarriers such as liposomes, micelles, silica, dendrimers, gold, and polymer NPs are often used in anticancer PDT for improving PS’s stability, delivery, tumour selectivity and accumulation through passive and active targeting, drug release, TME targeting (through hypoxia targeting and pH response) and deep PDT (through upconversion, two-photon PDT, self-illuminated PDT) and for including other functionalities (such as for imaging and combination therapies) [110]. Nanomaterials are considered in many PDT studies as they allow for different combinations of treatments that can overcome the limitations and disadvantages of the single-therapy treatments. In a comprehensive review by Chen et al., nanoparticles (NPs) for PDT and combinations of PDT with other therapies, most recently with photothermal therapy (PTT), were described, including gold NPs in PDT/PTT (using different wavelengths for activation of each process), silver NPs (with different morphologies adjustable for ^1^O_2_ production), silica NPs with encapsulated PSs (facilitating the delivery of hydrophobic PSs), quantum dots (QDs), upconversion nanoparticles (UCNPs), various carbon-based nanomaterials, different two-dimensional (2D) nanomaterials and nanoscale metal-organic frameworks (MOFs) [111]. Indocyanine green (ICG) is a clinically approved dye with maximum absorption in the NIR region (at ~800 nm) that is especially interesting for combination therapies because it can be used as a PS, for PTT and for fluorescence imaging [112]. For example, a recent report described NPs with ICG (for PTT) and chlorin e6 (Ce6) (for PDT), cisplatin for chemotherapy and a peptide (cRGD) and folate on the surface of the same NPs for active targeting [113]. The prepared NPs were tested on MCF-7 (human breast) and SGC-7901 (gastric cancer) cells, with PTT initiated by 5 min of irradiation at 808 nm (1.54 W/cm^2^), while PDT was initiated by 3 min of irradiation at 670 nm (1.0 W/cm^2^), and the obtained results showed improved drug delivery and synergistic anticancer effect [113]. Among the newer approaches to improving drug delivery is also “magnetic targeting”, which is based on exploiting magnetic field gradients and detection by magnetic resonance imaging (MRI) and uses iron oxide NPs in addition to a PS for PDT, imaging and other targeting agents [114,115,116,117]. 

However, even though nanotechnology offers many possibilities and solutions to current limitations of traditional PDT, there are still many challenges, most of them concerning biosafety, which needs to be thoroughly investigated. Furthermore, methods for the preparation and application of the NPs with PSs have yet to be standardised. Since there are now many different systems, some are overly complex and are thus difficult to study and compare, and only a small number of them are focused on optimisation and preclinical studies [110,111]. 

### 1.6. Combining PDT with PDT

The combination of two different PSs for PDT as a specific type of combination therapy (PDT/PDT), which is the main topic of this review, has some hypothesis-based approaches that are like the other previously mentioned combinations of therapies. For example, if a combination of two PSs achieves a synergistic effect, they can be used in lower doses, and this could reduce dark toxicity and help avoid skin photosensitivity in both anticancer and antimicrobial PDT. Furthermore, the use of two PSs to combine the targeting of different organelles leading to different pathways of cell death and to target tumour cells as well as tumour vasculature aims to increase the chances of overcoming cancer resistance mechanisms and of achieving a more complete response to PDT. Similarly, two different PSs in antimicrobial PDT could have better chances of avoiding possible resistance.

The first combination of two PSs was used for PDT against cancer, so it will be discussed in the next section, along with the main approaches and other examples that have followed thereafter. In recent years, there has been a significant increase in the number of articles focused on the study of upconversion nanoparticles. This approach appears to be particularly suitable for applications that involve two PSs in PDT, offering several special advantages. So, these results will be reviewed in a special section. 

Finally, there are significantly fewer combinations of two PSs for antimicrobial PDT, and they are described last. Mainly, the PubMed database was used to search for all articles describing the use of two PSs for PDT. We did not limit this search to any time period because we felt it was important to include the first examples whose results influenced the continuation of these studies, and because, as mentioned, it turned out that the examples were very unevenly distributed in time. The term single-PS PDT will be used when there is a comparison with the treatment using only one PS; however, dual-PDT, dual-PS PDT and similar terms will not be used for describing the application of two PSs in PDT to avoid confusion with any aforementioned applications where two distinct types of agents are used (e.g., one for PDT and one for PTT) or a single agent with two distinct roles is used in the treatment.

## 2. Combining Two PSs in PDT against Cancer

The first PS approved for anticancer PDT was Photofrin II. It was first approved in Canada (1993) for bladder cancer, then the following year in Japan for lung cancer, and then four years later in the US for oesophageal cancer [118]. The first combination of Photofrin II with another PS in a study for anticancer PDT was reported in 1990 [119]. Since Photofrin II has side effects, the most significant of which is high and long photosensitivity, the idea behind using it with another PS in combination was to reduce the individual amounts of each PS without reducing the overall effect of the PDT, the benefit being fewer side effects. Photofrin II was first combined with an anionic porphyrin, *meso*-tetra-(4-sulfonatophenyl)porphyrin (TPPS4) (Figure 2), and tested against EMT-6 mammary tumours in BALB/c mice [119]. Two different wavelengths were applied, with irradiation at 658 nm for Photofrin II and 630 nm for TPPS4, the light density for both being 100 mW/cm^2^. In the case of the single-PS PDT, the total dose was 60 J/cm^2^, while in the combination of the two PSs, it was 30 J/cm^2^ (Table 1). Moreover, the amount of the PS was 5 mg/kg in the single-PS PDT and 2.5 mg/kg of each in the combination PDT [119]. From the combination of the two PSs with two wavelengths, an enhanced PDT effect was achieved with a 100% cure rate as opposed to the single-PS PDT with a 50% cure rate for Photofrin alone and 60% for TPPS4 alone [119]. However, although a synergistic effect was observed, which was thought to be due to the different pharmacology of the two PSs and different mechanisms (Photofrin—vascular targeting; TPPS4—direct cell killing), the difference in mechanisms could not be observed, and it was mostly vascular damage that led to tumour eradication [119].

As already mentioned, a single-PS PDT can be used to target both tumour cells and tumour vasculature by combining different DLI intervals, and such studies have also been conducted with Photofrin [120]. However, Photofrin II is more efficient and used in vascular targeting, and it was combined in this role with 5-aminolevulinic acid (ALA) in a study of both vascular- and cellular-targeted PDT with two PSs on mice bearing human colon carcinoma (WiDr and KM20L2) [121]. In vitro, 168 μg/mL of ALA was combined with 0.02 or 0.1 μg/mL Photofrin, and in vivo, 250 mg/kg of ALA was administered with 1, 2.5 or 5 mg/kg of Photofrin, followed by 3 h of incubation and irradiation at 632 nm (with density 150 mW/cm^2^) for 15 min (resulting in a dose of 135 J/cm^2^); the same PS concentrations were also tested in the single-PS PDT treatments in vitro and in vivo [121]. An enhanced PDT effect against cancer cells was observed with the two PSs in vivo, with a significantly slower tumour growth compared to the untreated tumour and tumours treated with single-PS PDT [121]. Up to a point, the enhancement of the PDT effect was greater the higher the concentrations of Photofrin used in the combination studies, but most importantly, in all cases, the dose of Photofrin was low enough to avoid skin phototoxicity. Moreover, at such a low concentration, it did not interfere with PPIX generation from ALA [121]. It is worth noting that this is the first study of the kinetics of the uptake and clearance of ALA in tumour and of the accumulation and elimination of PPIX (when formed from ALA) in the plasma and tumour, which showed that short half-lives are associated with negligible risk of skin phototoxicity [121].

An important advancement in the research of PDT with two PSs was made when, in addition to the selection of two PSs with different targeting and wavelengths of light required for activation, the effect of the sequence of irradiation with two different wavelengths was examined. Benzoporphyrin derivative (BPD-MA), well-known for vascular targeting, and 5-ethylamino-9-diethylaminobenzo[a] phenothiazinium chloride (EtNBS) (Figure 2), which is associated with direct cell killing in PDT, were combined for in vivo PDT on BALB/c mice with subcutaneous EMT-6 murine sarcoma [122]. In the first approach, irradiation at 652 nm (for EtNBS) was followed by irradiation at 690 nm (for BPD-MA), while in the second approach, EtNBS was added 3 h after BPD-MA, and 3 h later, irradiation at 690 nm was applied, followed immediately by irradiation at 652 nm (with light density in all cases being 100 mW/cm^2^) [122]. Significantly better results were achieved with the first approach (a 95% tumour reduction), while BPD-MA was better than EtNBS in single-PS PDT. However, even after doubling the PS concentration and increasing light density, single-PS PDT was still much less effective than any combination of two PSs, confirming a synergistic effect for the latter [122]. It is worth emphasizing that in the single-PS PDT group in this study, 77% of mice died after two days, while there were no mice deaths after PDT with the two combined PSs, and it was suggested that the immune response played an important part in the success of the combination PDT [122]. 

The targeting of different organelles is another common approach in combining two PSs for anticancer PDT. In the first published study using this approach, performed on L12 10 leukemic cells, a haematoporphyrin derivative (HpD), which targets the cytoplasmic membrane, was combined with rhodamine 123 (Rh123) (Figure 2), a lipophilic cationic dye that targets mitochondria [123]. Incubation with HpD (at the concentrations of 0.5, 1 and 2.5 μg/mL) was conducted for 15, 30 or 45 min before and with Rh123 (at the same concentrations) 30 min before irradiations at 488 and 514 nm, with the light doses of 25 and 50 J/cm^2^, respectively. An increase in concentration and light dose was shown to decrease the surviving fraction of the control cells for both HpD and Rh123, while an increase in incubation time followed the same trend, but only for HpD [123]. Altogether, the results were not very satisfactory because an increase in phototoxicity by combining the two PSs could only be achieved when the highest concentration of HpD was used (2.5 μg/mL), while with 1 μg/mL of HpD, in most cases, the PDT effect was lower for the combination of the PSs than for Rh123 alone [123]. 

A much more promising targeting of different organelles in PDT was achieved by combining water-soluble *meso*-tetra-(4-*N*-methylpyridyl)porphyrin (TMPyP) and hydrophobic zinc(II) phthalocyanine (ZnPc) (Figure 2), which was incorporated in the liposome on HeLa cells that were irradiated after 1 h of PS incubation, then irradiated with red light at 650 nm (4 mW/cm^2^) for a final light dose of 2.4 J/cm^2^ (for 10 min irradiation) [124]. The two PSs were differently localised, with localisation of ZnPc in the Golgi apparatus and of TMPyP in lysosomes, and their photoactivation led to a significant and synergistic PDT effect [124]. These results prompted further research with the same pair of PSs, which were simultaneously administered in low concentrations (TMPyP—1 × 10^−6^ M in PBS; ZnPc—5 × 10^−8^ M in liposomes) to human cervix adenocarcinoma (HeLa), human keratinocyte (HaCaT) and human breast adenocarcinoma (MCF-7). After 1 h of incubation, two low doses of light were used for photoactivation (irradiation at 650 nm with density 4 mW/cm^2^ for 10 or 15 min), and the results were evaluated after 24 h or 48 h [125]. In single-PS experiments with the same PS concentrations, there was neither dark toxicity nor phototoxicity, while with two PSs applied together, there was no dark toxicity; however, after irradiation, there was a significant PDT effect, which proved synergistic, in all three cell lines in vitro [125]. It was shown that HeLa cells subjected to a low light dose (10 min irradiation) undergo apoptosis via an intrinsic (mitochondrial) pathway as soon as 3 h after PDT with two PSs. After 24 h, most of the cells were apoptotic, while with a higher dose of 3.6 J/cm^2^, most of them were necrotic [125]. In vivo, in C57BL/6 mice with amelanotic melanoma, the PSs (4.1 mg/kg TMPyP and/or 0.5 mg/kg ZnPc) selectively accumulated in the tumour, and after 24 h of incubation, the animals were treated with light at 600–700 nm (density 175 mW/cm^2^, dose 300 J/cm^2^). In comparison to the single-PS treatments, PDT with two PSs notably slowed down tumour growth and with the greatest efficiency [125].

David Kessel and his group have made a significant contribution to the research of PDT with two PSs by studying different organelle-targeting mechanisms and the effects of different activation sequences [126,127,128]. They hypothesised that the enhanced and synergistic PDT effect could be achieved by using low-dose PDT to first induce damage to lysosomes, followed by a low-dose PDT targeting mitochondria. They conducted a study with BPD-MA (mitochondria localising PS) and mono-L-aspartyl chlorin e6 (NPe6) (late endosomes and lysosomes localising PS) on murine hepatoma 1c1c7 cells to analyse how lysosomal photodamage enhances sequential mitochondrial photodamage [126]. After incubation of 0.5 μM BPD-MA and 40 μM of NPe6 for 1 h, the cells were irradiated at 690 nm (for BPD-MA) and at 660 nm (for NPe6) in two different sequences [126]. Diethyl-3-30-(9,10-anthracenediyl) bis acrylate (DADB) was used to evaluate ^1^O_2_ production, while aminophenyl fluorescein (APF) was used for hydroxyl radical evaluation (for both dyes fluorescence is quenched upon reaction with ROS), and the measurements confirmed that the main type of ROS in PDT for both PSs is ^1^O_2_, and that BPD-MA is a higher producer of ^1^O_2_; however, there was no higher production of ^1^O_2_ in any of the studied sequential PDT protocols [126]. On the other hand, clonogenicity was significantly reduced (down to 17%) when NPe6 was activated first, as opposed to BPD-MA being applied first and followed byNPe6 (down to 58%), and when lysosomes were damaged by PDT first, the loss of the mitochondrial membrane potential was significantly potentiated [126]. Low-dose lysosomal PDT with NPe6 did not change the localisation of BDP-MA and did not increase hydroxyl radical production, and although the mechanism of potentiation of the PDT effect was not completely clear, it seems that the sequence where NPe6 is activated before BPD-MA amplifies pro-apoptotic signalling [126]. The advantages of the sequential protocol in which lysosomes are targeted first were confirmed in the continuation of the research in which Photofrin was introduced in addition to BDP-MA and NPe6. Three combinations were tested—the activation of NPe6 followed by BPD-MA, the activation of NPe6 followed by Photofrin, and the activation of Photofrin followed by BPD-MA [127]. Photofrin was incubated for 16 h and activated at 630 nm; its absorption proved to be sufficiently distinct such that, in all three cases, it was possible to activate each PS sequentially without activating the other at the same time. Photofrin successfully played both roles, inflicting lysosomal damage when photoactivated before BPD-MA and mitochondrial damage when activated after NPe6 [127]. Similar studies on the human non-small-cell lung cancer A549 cell line showed that A549 cells were less sensitive to PDT with a sequential protocol than 1c1c7 cells, but still, a synergistic PDT effect was observed, and DEVDase activity was increased in both cell lines. However, the loss of the mitochondrial membrane potential was substantial only in 1c1c7 cells [128]. Interestingly, it was found that paraptosis contributed to PDT killing, and this is one of the first studies where paraptosis, a type of programmed cell death that is caspase-independent and morphologically different from apoptosis and necrosis, characterised by the formation of vacuoles from the ER, is linked to PDT [128].

A synergistic effect of combining two specific PSs in anticancer PDT was discovered incidentally during a clinical trial when a patient treated with ALA for a breast tumour developed a phototoxic reaction due to hypericin (HYP) taken as an herbal antidepressant [129]. Their activity together was then tested on keratinocyte HaCaT cells that were incubated with HYP for 24 h and/or ALA for 3 h, and after irradiation (at 400 ± 780 nm), a synergistic effect was confirmed [129]. A few years later in a similar study, 0.5 mM ALA was combined with 60 nM HYP, and after 4 h of incubation in human endometrial cancer cells (HEC-1A), these cells were irradiated with red light at 635 nm or with white, non-coherent light (at 400–800 nm), which includes 590 nm, the maximum absorption for HYP, for 24 min (reaching a dose of 2.5 J/cm^2^), and single-PS PDT experiments were conducted for comparison [130]. The highest PDT effect (a 45% reduction in control cells) was reported for the combination of the two PSs and white light, and it was observed that there was more PPIX produced from ALA in the presence of HYP than without it and that HYP increased the PDT effect of PPIX [130].

The aforementioned research has encouraged the use of HYP in new combinations, such as with *meso*-tetrahydroxyphenylchlorin (mTHPC) (Figure 2), a PS that has significant toxicity in the dark; the main goal of using this combination is thus to reduce its dark toxicity [131,132]. This was indeed achieved on head and neck squamous cell carcinoma cell lines (HNSCC cell lines UMB-SCC 745 and 969), using HYP and mTHPC together in a 1:1 ratio. This was especially evident for HYP, which was significantly reducing cell viability in all tested concentrations (from 0.6 to 10 μg/mL), but also for mTHPC, which was toxic above 5 μg/mL. In their mixture, there was no significant toxicity up to a concentration of 5 μg/mL of each PS [131]. It was postulated that HYP interacted with the liposomal formulation of mTHPC, since the localisation of HYP in cells is usually associated with membranes, and its fluorescence was also observed to be reduced [131]. Furthermore, it was shown that there is no mutual quenching and that the mixture even provides mTHPC a longer photostability, which would otherwise be much shorter than that of HYP (HYP was stable for at least 24 h, while mTHPC was only stable for 6 h; in the mixture, mTHPC was still present after 8 h) [132]. The two PSs differ not only in their localisation, but also in the type of cell death they trigger, so unlike HYP, which is localised in membranes and is associated with apoptotic cell death, mTHPC is not specific to any organelle but is diffusely distributed in cells and is associated with necrotic cell death [132]. For PDT, a mixture of 1.25 μg/mL of each PS was irradiated, after 5 h of incubation, with white light (32 mW/cm^2^) for 1 min, and this led to a more than 90% reduction in cell viability, which was similar to HYP-PDT (with a 2.5 μg/mL HYP concentration) and more efficient than for mTHPC-PDT (83% for both cell lines) [132]. Although both types of cell death were observed, the morphological changes in the cells were more similar to those for HYP-PDT [132]. Interestingly, studies of ^1^O_2_ by direct detection at 1270 nm (HYP was excited at 590 nm and mTHPC at 415 nm) showed that the observed synergistic effect in PDT did not come from more ^1^O_2_ being produced, since measured ^1^O_2_ yields for HYP, mTHPC and their 1:1 mixture were 0.25, 0.66 and 0.4, respectively [132]. Other ROS were also measured using CM-H2DCFDA, whose oxidation product is fluorescent, and a stronger fluorescence was reported after HYP-PDT in both cell lines, especially in UMB-SCC 745. Meanwhile, after mTHPC-PDT, only weak fluorescence was observed in UMB-SCC 969, and after PDT with a 1:1 mixture of HYP and mTHPC, the fluorescence was similar to HYP-PDT, although slightly stronger in UMB-SCC 969 [132]. In all PDT experiments, whether with either PS or with a mixture of both, the heat shock 70 kDa protein 6 (HSPA6) was highly expressed [132]. 

Among distinct types of cancer, melanoma continues to be one of the biggest challenges for PDT due to melanin and different resistance mechanisms, so different combination therapies are being investigated, including PDT with two PSs [133]. In one such study, Photodithazine (PDZ) (Figure 2), which is used for cellular-PDT, was combined with BPD-MA (for vascular-PDT) on a mouse model with pigmented cutaneous melanoma [134]. Photodithazine (1.0 mg/kg for single-PS PDT, 0.5 mg/kg for its combination with BPD-MA) activated before its combination with BPD-MA, was activated after a 60 min incubation by irradiation at 670 nm (density 100 mW/cm^2^) for 1000 s (dose 100 J/cm^2^, in combination 60 J/cm^2^). BPD-MA (0.8 mg/kg for single-PS PDT, 0.4 mg/kg for combination with PDZ) was incubated for 15 min and irradiated at 690 nm (80 mW/cm^2^) for 1000 s (80 J/cm^2^, in combination 40 J/cm^2^), and 1,2-propanediol was also included as an optical clearing agent. All of these together in combination led to the first complete eradication of pigmented melanoma [134]. Uveal melanoma is an exceedingly rare but lethal malignancy [135], and PDT with two PSs as a possible treatment was tested against the uveal melanoma cell line (C918) in vitro in 2-D, combining ALA (1 mM for single-PS, 100 μM for combination) with (5,10,15,20-tetrakis-(*N*-methylpyridynium-4-yl)porphyrin)palladium(II) (Pd(T4)) (Figure 2) (10 μM for single-PS, 2.5 μM for combination) [136]. The cells were incubated for 2 h, then irradiated at 405 nm (60 mW/cm^2^) for 88 s (5 J/cm^2^), and even though in combination sub-optimal concentrations of the PSs were used, PDT resulted in a 52.8% cell viability as opposed to the single-PS PDT that resulted in 97.1% (ALA) and 78.2% (Pd(T4)) viability [136].

Glioblastoma, a type of brain tumour, is another difficult-to-treat aggressive type of cancer, for which PDT is being investigated as a possible therapy. In fact, PPIX derived from ALA is approved by the FDA for fluorescence diagnostics (FD) and visualisation during surgery [137,138]. In one male patient with recurrent glioblastoma after surgical removal, ALA-PDT was used, and for the second surgery, 20 mg/kg of ALA was administered 4 h before FD, followed by irradiation at 635 nm for 6.5 min (30 J/cm^2^) for PDT [139]. After 4 months and another recurrence of glioblastoma, in addition to ALA (20 mg/kg, incubation 4 h before FD) during the third surgery, Ce6 (1 mg/kg in 100 mL saline, 3 h before FD) was also administered [139]. The two PSs together improved the quality of FD during the surgical removal of the glioma, with Ce6 being mostly accumulated in the tumour vascular system and somewhat in tumour cells (lysosomes and mitochondria), whereas PPIX accumulated in tumour tissues. After resection and PDT at 660 nm for 14 min (60 J/cm^2^), subsequent FD did not show any fluorescence in the tumour bed [139]. 

Table 1 summarises the combinations of two PSs in anticancer PDT.

**Table 1 pharmaceuticals-16-00613-t001:** Summary of the combinations of two PSs in PDT against cancer with representative examples.

Photosensitisers (Targeting—If Any)	Tumour Type/Cancer Cell Line	Irradiation Wavelength (Fluence Rate; Dose of Light)	Synergistic Effect	Other Observations/Outcomes	Ref.
Photofrin (vascular) TPPS4 (cellular)	EMT-6 mammary tumour (in vivo)	658 nm (Photofrin) (100 mW/cm^2^) and 630 nm (TPPS4) (100 mW/cm^2^); light dose together 60–80 J/cm^2^)	Yes/100% cure	Mostly vascular damage; less side effects	[119]
Photofrin (vascular) ALA (cellular)	WiDr and KM20L2 human colon carcinoma (in vivo and in vitro)	632 nm (150 mW/cm^2^; 135 J/cm^2^)	Enhanced PDT effect in vivo, but not in vitro	No skin phototoxicity	[121]
BPD-MA (vascular) EtNBS (cellular)	EMT-6 murine sarcoma (in vivo)	Sequential: (1) 652 nm (EtNBS) (100 mW/cm^2^), (2) 690 nm (BPD-MA); (100 mW/cm^2^)	Yes	No mice death; immune response	[122]
HpD (cell membrane) Rh123 (mitochondria)	L12 10 leukemic cells (in vitro)	488 nm (HpD) (25 J/cm^2^) and 514 nm (Rh123) (50 J/cm^2^)	No		[123]
TMPyP (lysosome) Liposomal ZnPc (Golgi)	HeLa, HaCaT, MCF-7 cells (in vitro) and melanoma (in vivo)	650 nm (4 mW/cm^2^; 2.4 J/cm^2^) and 3.6 J/cm^2^ in vitro) and 600–700 nm (175 mW/cm^2^; 300 J/cm^2^ in vivo)	Yes	Apoptosis; tumour growth retardation	[125]
BPD-MA (mitochondria) NPe6 (lysosomes)	Murine hepatoma 1c1c7 cells (in vitro)	Sequential: (1) 660 nm (NPe6), (2) 690 nm (BPD-MA)	Yes (but not increased ROS)	Amplified pro-apoptotic signalling, reduced clonogenicity	[126]
HYP ALA	HEC-1A human endometrial cancer cells (in vitro)	White non-coherent light: 400–800 nm (2.5 J/cm^2^)	Yes	HYP increased the PDT effect of PPIX	[130]
HYP (cell membrane—apoptosis) Liposomal mTHPC (diffusely distributed in cells—necrosis)	UMB-SCC 745 and 969 head and neck squamous cell carcinoma (in vitro)	White light (32 mW/cm^2^)	Yes (but not increased ROS)	Reduced dark toxicity, longer photostability of mTHPC, prevails apoptosis (from HYP-PDT)	[132]
BPD-MA (vascular) PDZ (cellular)	(Non-)Pigmented cutaneous melanoma (in vivo)	Sequential: (1) 670 nm (PDZ) (100 mW/cm^2^; 60 J/cm^2^), (2) 690 nm (BPD-MA) (80 mW/cm^2^; 40 J/cm^2^)	The first complete eradication of pigmented melanoma	Pigmented melanoma best response with optical clearing	[134]

### Conjugates and Nanocarriers for PS Delivery and Imaging

The conjugation of two PSs is sometimes used for facilitating the delivery of each PS and/or for enabling an activatable release at the site of PDT action. It has already been mentioned that a liposomal formulation is often used to deliver hydrophobic ZnPc for anticancer PDT applications [140], and how ZnPc in liposome was combined with TMPyP for targeting different organelles has also been described [125]. Another interesting approach describes the conjugation of ZnPc to four molecules of ALA with the aim of improving the solubility of ZnPc while simultaneously hiding the hydrophilicity of ALA [141]. The conjugate was tested on HeLa cells, and it was shown that the ZnPc-ALA conjugate enters the cells and then releases ZnPc and ALA to form PPIX, so irradiation with blue (at 408 nm) and red (at 640 nm) light from LED sources were then used to activate both PSs at the same time after 3 h or 24 h of incubation [141]. PPIX fluorescence was the same after 3 h and 24 h, but the PDT’s efficacy was higher after 24 h of incubation. A comparison of the ZnPc-ALA conjugate with the PDT action of unconjugated ZnPc and ALA separately showed that the PDT activity of the conjugate was synergistic [141]. 

A recent paper describes a conjugate of two PSs targeting different organelles and producing different types of ROS. The PSs are released after hydrolysis catalysed by γ-glutamyl transpeptidase (gamma-glutamyl transferase, GGT), an enzyme that is overexpressed in multiple tumours [142]. Pyropheophorbide a (PPa) (Figure 2), which accumulates in mitochondria, and EtNBS (which localises in lysosomes) were covalently attached to a γ-glutaminyl moiety, and in the prepared probe, which has strong absorption at 660 nm, both PSs quenched each other; however, after their release and upon irradiation, they were allowed to produce ^1^O_2_ (PPa) and peroxyl radical anion, O_2_^•−^ (EtNBS) [142]. The probe was tested in vitro on human hepatocyte carcinoma (HepG2) and human glioblastoma (U87) cells and after incubating 0.4 μM of the probe for 4 h and irradiating it at 660 nm (40 mW/cm^2^) for 15 min, the PSs were released upon hydrolysis by GGT and organelle selection, resulting in apoptosis prevailing over necrosis [142]. The probe was also tested in vivo, on Balb/c mice with U87 cells, and after 6 h of incubation of 100 nmol of the probe, followed by irradiation at 660 nm (300 mW/cm^2^) for 15 min, a slowing of tumour growth was observed [142].

As previously mentioned, there are numerous examples that describe the use of two PSs for the targeting of different organelles and/or for combining vascular-PDT with cellular-PDT. However, there are also a few examples where the same PS was used in different formulations to exploit different localisations, such as BPD-MA being used as a well-known vascular-targeting drug for age-related macular degeneration [143], registered under the name Visudyne (targeting mitochondria and ER), and as a phosholipid-anchored BPD-MA liposome that enters the cell through the endosome and targets lysosome [144]. To test PDT with these two formulations, 3D OVCAR5 (ovarian cancer cell line) nodules were incubated with lipid-anchored BPD-MA for 24 h; then, Visudyne was added for another 1.5 h, followed by irradiation at 690 nm (50 mW/cm^2^) for up to 20 min [144]. This combination of Visudyne and BPD-MA liposomes in PDT resulted in a considerable reduction in tumour area and an enhanced PDT effect compared to either of the single PS formulations, using comparable concentrations and the same or lower energy densities [144]. The same study was performed on monolayer OVCAR5 cultures, with a light dose of 200 mJ/cm^2^, and in both studies, simultaneous targeting of lysosomes, mitochondria and ER, similar to sequential targeting (where lysosomes are targeted before mitochondria), confirmed that calcium ions released from damaged lysosomes enhance parapoptosis after mitochondrial/ER damage, resulting in improved PDT [145]. The authors suggest that the use of the same PS in two different (targeting) formulations irradiated at one wavelength for activation should allow for simpler PDT protocols, as opposed to PDT with two different PSs activated with different wavelengths, while still benefiting from lower PS concentrations and light doses [145].

Except for upconversion nanoparticles (UCNPs), which will be discussed in the next section, there are only few published reports describing nanocarriers used for the joint delivery of two PSs for anticancer PDT. In one example, PPIX and HYP were encapsulated together in lipid nanocapsules (LNC25) to improve the solubility of HYP and reduce dark toxicity and were tested on HeLa and MDA-MB-231 (human breast cancer) cell lines in vitro and in Swiss nude mice [146]. The nanocapsules ranged in size from 27 to 31 nm and weight percentages of 0.04% HYP and 0.045% PPIX for single therapies and of 0.02% HYP and 0.022% PPIX in LNC25s for a combination PDT; in all cases, the loaded LNC25 were incubated for 8 h and irradiated (400–700 nm) for 12 min [146]. Singlet oxygen quantum yield measurements (by *p*-nitroso-dimethylaniline method with Rose Bengal (RB) as a standard PS) showed that LNC25 encapsulation significantly improved ^1^O_2_ production for both PSs (under blue, red and visible light) compared to free PSs, which was explained by their reduced aggregation, while PPIX measured the highest production yield of ^1^O_2_ in comparison to the same concentration of HYP in LNC25 or both PSs (HYP + PPIX) in LNC25 [146]. The intracellular localisation of the loaded LNC25 in plasma and ER was reported, along with a significantly higher phototoxicity, described as a synergistic effect of combining the two PSs together (two times higher than with a single PS in LNC25) on both cell lines [146]. The in vivo distribution of loaded LNC25 studied 2 h after different types of injections showed that the intratumoural treatment was the best one for administration, and the PDT using two PSs encapsulated in LNC25 slowed down tumour growth more than both single-PSs in LNC25 [146].

Since their discovery, boron dipyrromethene (BODIPY) dyes have attracted a great deal of attention because of their spectroscopic properties that make them excellent fluorophores, and their development has been largely related to their diagnostic applications [147]. However, syntheses and modifications in the structure were developed very quickly, which enabled them to be used as PSs for PDT. Together with their fluorescent properties that can be sufficiently retained, they become very interesting as possible theranostic agents [148,149]. The most common approach to improving ISC and ^1^O_2_ production is halogenation at the 2- and 6-positions, and although no BODIPY molecules have yet been approved for clinical trials, research into opening areas of application is continuing [150,151]. To improve the properties of both PDT and imaging, in some instances, BODIPY was prepared in a conjugate with porphyrin [152], or more commonly with phthalocyanines [153], and via the assembly of Pcs with conjugates of BODIPY and NPs such as nanodiamonds [154] and graphene quantum dots [155]. However, these conjugates are all constructed as a single PS and not much biological activity has been described. In addition, one conjugate of BODIPY, thus a single PS, but with an observed synergistic effect in vivo and high potential for fluorescence imaging, is the one with diketopyrrolopyrrole (DPP) in NPs [156]. Two BODIPY molecules were used as donors at both ends of the DPP, which was used as an acceptor, while iodine atoms were introduced in the BODIPY to increase ISC and ^1^O_2_ production. Although this expectedly lowered the fluorescence of the BODIPY’s moiety, fluorescence was increased by the DPP [156]. A low dark toxicity and high synergistic PDT effect on HeLa cells was shown by NPs with the conjugate, with an 80% increase in ^1^O_2_ quantum yield and a 5% increase in the fluorescence quantum yield (compared to unconjugated BODIPY and DPP), both in vitro and in mice where fluorescence proved to be the strongest after 4 h of incubation and remained for 24 h [156]. Furthermore, there was no visible damage to the main organs, and the tumour was effectively destroyed at low PDT doses [156].

To conclude this section on the use of two PSs in anticancer PDT, the main approaches and the main advantages over PDT with one PS can be summarised as shown schematically in Figure 3.

## 3. Upconversion Nanoparticles with Two PSs

Upconversion nanoparticles based on lanthanides were introduced in the last years of the 20th century, which curiously coincided with the registration of the first photosensitiser for PDT [157]. Upconversion nanoparticles (UCNPs) are inorganic nanocrystals, most commonly with α- and β-NaYF_4_ (but also NaGdF_4_ and NaLuF_4_), used as host lattices and doped with lanthanide ions that have excited states with long lifetimes due to sharp 4f–4f transitions, which allows for the sequential absorption of two or more photons [157,158,159]. Consequently, this results in the conversion of two or more photons of lower energy (NIR), via a nonlinear optical process, into one photon of high energy (UV and visible) [158]. In the context of PDT, the two NIR wavelengths that are most often used for absorption by UCNPs at 810 and 980 nm are deeply penetrating, but the photons do not have enough energy for ^1^O_2_ production. However, the upconversion emission occurs in the part of the EMS where PSs usually absorb strongly, and moreover, there is the possibility of using two or even three PSs of different absorption wavelengths together with UCNPs since upconversion usually results in two or three different emission maxima (e.g., blue, green and red) [157,158,159]. 

The two main transition processes that are employed with UCNPs are excited state excitation (ESA) and energy transfer upconversion (ETU), but cooperative sensitisation upconversion (CSU), photon avalanche (PA) and cross relaxation (CR) are also used [157,158]. ESA is based on a single lanthanide ion (Er^3+^, Ho^3+^, Tm^3+^, Nd^3+^) whose photon absorption leads to the intermediate excited state E1, where, due to its long lifetime, the absorption of another photon to the excited state of higher energy also occurs. ETU involves two nearby ions: the first one is a sensitiser ion (commonly Yb^3+^ or Nd^3+^) excited to E1, from where it then transfers its energy (and relaxes back to its ground-state G) to the G and E1 state of the second ion, which is an activator (Er^3+^, Ho^3+^, Tm^3+^) that is then excited to the state from which emission occurs [157,158,159]. For PDT, energy is then transferred to the PS, usually by FRET, and thus upconversion emission should correspond to the maximum absorption of the PS. The PS also needs to be close to the ions. As opposed to two-photon excitation that is also used for deeper tissue penetration, but which requires expensive lasers and often involves PSs involved of low photochemical stability, UCNPs in PDT have advantages such as higher chemical stability, low background signals, large anti-Stokes processes and narrow emission bands, and photoactivation can be achieved over larger areas with lower energy irradiation produced by more affordable lasers [158,160,161]. 

In planning the synthesis of UCNPs (extensively reviewed by Zhu et al. [162]), the size of the particles should be considered to ensure an optimal surface-to-volume ratio, and the doping concentration must be adjusted to avoid quenching by cross-relaxation and energy migration due to higher doping concentrations [163,164]. Since weak absorption and a narrow absorption band are some of the most serious limitations of UCNPs, solutions are being sought that would enable higher doping concentrations and increased absorption while avoiding quenching. Most commonly, an inert shell is used to protect UCNPs with high doping concentrations from surface quenchers, and other strategies include sensitisation with dyes, using Yb^3+^ and Nd^3+^ as activators, a large unit cell of the host crystal, a better distribution of the ions used for doping and applying a higher power density for excitation [163,164,165]. In addition to PDT, UCNPs can be used for other biomedical applications, such as for bioimaging or in combination with chemotherapy, radiotherapy, gene therapy and PTT [165,166,167]. For these applications, UCNPs can be functionalised with groups for conjugation to biomolecules to introduce the hydrophilicity and biocompatibility needed for tumour imaging, antimicrobial detection and for medical treatments. However, the biodistribution of such functionalised UCNPs prepared so far has not been investigated well in vivo in humans [167]. 

Thanks to their activation by deeper-penetrating NIR light, UCNPs have an enormous potential for use in PDT, especially for deep tissues and solid and large tumours [165]. Usually, a PS (Figure 4) is loaded non-covalently by encapsulating it in silica or by physical adsorption. Alternatively, it can be covalently conjugated to UCNPs, and often active tumour targeting and imaging modalities are also added [157,160,165,168]. The most common limitations of UCNPs for PDT identified so far have been the low PS loading and the low yield of ^1^O_2_ production; thus, approaches using two PSs are increasingly being explored, which have so far shown greater efficacy in comparison to single-PS PDT (Figure 5) [158,165].

In the first example of an in vivo application of UCNPs in PDT, the nanoparticles were loaded with two PSs, merocyanine 540 (MC540) and ZnPc, and were conjugated with folic acid for tumour targeting and PEG [169]. In this research, UCNPs were used to treat C57BL/6 mice with melanoma, and after 4 h of incubation, irradiation with 415 mW/cm^2^ at 980 nm was applied, leading to upconversion emission in the green (~540 nm) and red (~660 nm) parts of the EMS, which corresponds to the absorptions of MC540 and ZnPc, respectively, and consequently led to their photoactivation [169]. Another dose was given 5 days later, and the complete treatment resulted in reduced tumour growth. Meanwhile, PDT on B16-F0 melanoma cells in vitro (40 min of irradiation at 2.5 W/cm^2^) showed decreased fluorescence of 9,10-anthracenediyl-bis (methylene)dimalonic acid (ABDA) after 20 min of irradiation, confirming ^1^O_2_ production [169]. In a recent example with the same two PSs, UCNPs were constructed as the core–shell model (NaYF_4_:Yb/Er/Nd@NaYF_4_:Nd) for 808 nm laser irradiation to reduce water absorbance and consequently the water heating effect, which is a well-known problem with irradiation at 980 nm [170]. Furthermore, energy transfer is not very efficient when all the lanthanide ions are in the same matrix, so the core–shell structure allows for high doping, increases upconversion efficiency and also improves the stability of UCNPs in water and biological media [170,171]. In such a design, Nd^3+^ as a sensitiser transfers energy to Yb^3+^; then, the energy is transferred to the ion responsible for emission (Er^3+^) [170]. Furthermore, folic acid was used for active targeting since NPs in general cannot exploit enhanced permeability and retention (EPR) effects for tumour targeting [170,171]. When loaded with two PSs, UCNPs produced more ^1^O_2_ and ROS in vitro than UCNPs loaded with either of these two PSs, resulting in more effective PDT against HeLa cells, and active targeting was proven both in vitro and in vivo in Kunming mice bearing H22 (hepatocellular) tumours [170].

It has been reported that an intensive white light with three maxima (at 475 nm, 540 nm and 650 nm) is emitted upon irradiation at 980 nm by UCNPs whose cores are made of a NaYF_4_ host lattice doped with Yb^3+^ as a sensitiser, Er^3+^ as an activator and Tm^3+^as a co-activator (gives blue emission) and whose shells are made of mesoporous silica with two PSs—ZnPc loaded by adsorption (to exploit red emission) and RB attached covalently (to exploit green emission) in addition to folic acid covalently attached to enhance cellular uptake [172]. To test in vitro PDT, HeLa cells were treated for 48 h with UCNPs with either no PS, one PS or two PSs. Compared to the lack of toxicity in dark conditions, after irradiation at 980 nm (2.5 W/cm^2^) for 20 min, significant apoptosis was detected by flow cytometry [172]. The bioimaging potential for photodynamic diagnosis (PDD) with white light using an optical microscope was demonstrated on human mesenchymal stem cells (hMSC) treated with the UCNPs and upon activation with 2.5 W/cm^2^ at 980 nm [172]. Furthermore, it was shown that they could also be probed under fluorescence microscope simultaneously with NIR and UV light (with the stain Hoechst for nucleus giving blue fluorescence) [172]. The same two PSs, ZnPc and RB, were employed in the core–shell UCNPs (NaYF_4_:20%Yb^3+^,2% Er^3+^@NaYF_4_) with poly(allylamine) (PAAm), for PDT in vitro on A549 cells and in vivo on mice injected with Hepa1–6 cells (murine model of hepatocellular carcinoma) [173]. The lowest cell survival percentage (20%) was achieved in vitro after UCNPs (100 μg/mL) were incubated for 24 h and upon irradiation at 980 nm (0.25 W/cm^2^ for 15 min with 3 min stops between 5 min of irradiation). In vivo, a reduced tumour growth, with no damage to the main organs, was achieved with two PSs in UCNPs (17 mg/kg) and a light dose of 225 J/cm^2^ [173]. 

To overcome the low energy transfer from NPs to PSs and low ROS generation, a “sandwich-structure” of UCNPs (NaYF_4_@NaYF_4_:Yb,Tm/Ho@NaYF_4_) was designed, with a thin silica layer as a core and the two PSs, MB and RB, contained in the middle layer [174]. The mitochondria-targeting group (triphenylphosphine, TPP) was inside the silica layer, all the active ions were mostly in the middle layer (Tm^3+^ on Ho^3+^ were added to improve red emission) and the polyethylene glycol (PEG) layer was on the outside of the UCNPs and contained folic acid for tumour targeting [174]. These UCNPs have been shown to enter cancer cells via endocytosis, and upon illumination with a 980 nm laser, upconversion results in green (~540 nm) and red emission (~650 nm), corresponding to the absorption of RB and MB, respectively, subsequently leading to the apoptosis of MCF-7 cancer cells in vitro and to tumour reduction after in vivo PDT in a Balb/c nude mouse [174].

The combination of Ce6, intended for the absorption of the blue and red emissions, and RB, for the green part of the upconversion emission, was used for PDT via UCNPs with a core–shell structure (NaYF_4_:Yb,Er,Nd@NaYF_4_:Yb,Nd) and PEG-phospholipids (DSPE-PEG) employed for encapsulation, which is excitable at 808 nm and has emission peaks at 407 nm, 520, 539 and 653 nm [175]. Using ABDA, a significantly higher ROS production was measured from UCNPs encapsulating Ce6 and RB compared to UCNPs with only one or no PSs, and highly efficient cellular ROS and PDT were confirmed on B16BL6 melanoma cells; however, dark toxicity (38.5 μg/mL) was also observed due to the instability of UCNPs and the consequent leaking of Ce6 [175]. Similarly, a much higher ROS production compared to single-PS PDT was measured with the same combination of Ce6 and RB in UCNPs that consisted of a nanocore (NaYF_4_:Yb13.6%Er2.6%) and three nanoshells (NaYF_4_:Yb_n_, *n* = 5.8, 6.1 and 4.7%) [171]. Around 100 Ce6 and 50 RB molecules were linked to one UCNP, and these UCNPs, activatable at both 808 and 980 nm, were shown to localise with lysosomes in HeLa cells in vitro and cause mitochondrial damage upon photoactivation [171]. In the third example of the same pair of PSs, RB was non-covalently loaded by encapsulation in the silica layer, while Ce6 was conjugated (through the amide bond) on the outside, and the UCNPs (prepared as core–shell LiYbF_4_:Er@LiGdF_4_) were PEGylated in the end [176]. Different amounts of Er^3+^ ions as the active dopant allowing for excitation at 1550 nm were tested (10, 30, 50 and 70 mol%), and the UCNPs showed emissions at 550, 670 and 800 nm, although with higher concentrations, quenching of green and red emissions was also observed [176]. Finally, PDT was evaluated in vitro against Panc-1 (human pancreatic) cancer cells and in vivo on pancreatic tumour-bearing mice. To avoid overheating due to water absorption, irradiation (0.5 W/cm^2^) was conducted for 0.5 min, then stopped for 2.5 min, and this was repeated in cycles ten times [176]. Again, more ^1^O_2_ was produced with UCNPs with the two PSs, and the PDT was more efficient compared to single-PS loaded UCNPs; however, dark toxicity was also higher in the case of the UCNPs with the combined PSs [176]. The tissue penetration of laser irradiation, at 808 and 1550 nm, with different power densities was also investigated and compared by measuring energy intensities after penetration through tissue samples of different thickness (1–5 mm), and the measurements proved that the longer wavelength penetrates deeper [176].

In addition to PDT in vitro on HeLa cells and in vivo on tumour-bearing mice (U14 cells), three modes for imaging (UC luminescence microscopy, CT and MRI) were evaluated using UCNPs with an oleic-acid capped core (NaGdF_4_:Yb,Er) and a shell (NaGdF_4_:Nd,Yb) structure designed to separate Nd^3+^ from its activators to avoid energy back-transfer and subsequent quenching [177]. The two PSs combined were MC540 (negatively charged, attached through electrostatic interaction) and Ce6 (attached covalently into silica), and IR-783 commercial dye was used to prepare IR-808 dye as an antenna on the surface of the UCNPs for 808 nm absorption [177]. 

Titanium dioxide (TiO_2_), used as a shell layer of the UCNPs 25 nm in size and with a NaFY_4_:Yb^3+^, Tm^3+^ and Er^3+^ core, was combined with Ce6, which was coupled through amide bonds to TAT peptides used for nuclear targeting against multidrug-resistant cancer [178]. Upon the laser irradiation of 0.1 mg/mL of UCNPs at 980 nm (0.5, 1 and 2 W/cm^2^ for 5 min), their emissions at 362 and 655 nm activated TiO_2_ and Ce6, which led to DNA damage in MFC-7 and MCF-7/Dox cells [178]. The same cells were injected in mice for tumour growth, and only UCNPs with both PSs resulted in successful in vivo PDT that resulted in size reductions in both types of tumours, without signs of systemic toxicity [178]. TiO_2_ as a UV light absorber was also paired with another PS—hypocrellin A (HA, blue light absorption)—in core–multi-shell UCNPs in addition to hyaluronic acid used for tumour targeting [179]. PDT using these UCNPs proved efficient against HeLa cells in vitro and in Balb/c nude mice with HeLa tumours after irradiation by laser at 808 nm (2 W/cm^2^) [179]. 

Finally, a polymer based on fluorine and benzothiadiazole (PFSBT) was used in UCNPs as a blue light absorber for ^1^O_2_ production, and titanocene (Tc) within an apo-transferrin complex was used as a second PS, activatable at 345 and 361 nm for the generation of peroxyl radicals against cancer cells [180]. After irradiation at 980 nm, both types of ROS were detected in MFC-7 cells in vitro and in ICR mice with H22 tumour, and PDT led to a decrease in tumour volume, without observable changes in body weight [180]. 

Figure 5 summarises the main principle behind using two PSs in UCNPs for anticancer PDT and imaging.

## 4. Combining Two PSs in Photodynamic Antimicrobial Chemotherapy

Although Oskar Raab, in his famous experiment, discovered the photodynamic effect on a microorganism more than 120 years ago [181], PDT research and its possible applications in the control of microbial pathogens have only intensified in recent years. It is easy to understand how the discovery and development of antibiotics and other antimicrobial agents pushed antimicrobial PDT aside during the 20th century, but the emergence of antimicrobial resistance, especially antibiotic resistance, has brought antimicrobial PDT research back to the forefront [182,183]. One of the main advantages of antimicrobial PDT, often abbreviated as APDT or aPDT, but better known as photodynamic antimicrobial chemotherapy (PACT) or photodynamic inactivation (PDI) [184], is that the probability of developing resistance is extremely low due to its multiple targets and action sites, and PACT has already been shown in numerous examples to be effective against antibiotic-resistant bacteria [185]. However, as expected, there are far fewer examples of the use of two PSs in PACT (Figure 6) than in anticancer PDT, and one reason may be, as mentioned, that PACT began to develop later and more slowly than anticancer PDT. Another reason is that with a single-PS PACT, resistance to therapy has generally not been observed so far, thus reducing the need for the second PS [58].

Curiously, one of the first examples in literature of the use of two PSs for PACT seem to be initiated by the observed resistance to erythromycin after PACT with HpD against *Staphylococcus aureus* [186]. The combination of HpD with mTHPC was therefore used against wild-type *S. aureus,* and both were excited with white light. The results from this PACT indicated an additive effect, but a certain amount of dark toxicity was also reported for mTHPC [186]. The dark toxicity of mTHPC was also reported in another study published the same year (1999), in which hypericin (HYP) and Photofrin II were also included in the combinations tested for PACT against *S. aureus* in addition to mTHPC [187]. White light was used again for irradiation, in a dose of 100 J/cm^2^ (fluence rate 60 mW/cm^2^ for 28 min), and the mixture of two PSs was applied with half the concentration of each PS compared to the single-PS PACT (Table 2). The combination of mTHPC with Photofrin II, which did not show dark toxicity, upon photoexcitation led to their average values in PACT, thus there was no synergistic effect [187]. When all three PSs were combined, in concentrations that were one-third of the amount used for each in the single-PS PACT, bacterial growth was stimulated. It was suggested that HYP, which alone stimulated bacterial growth in conditions with and without light, had an antagonistic effect in combination with porphyrins and thus inhibited their PACT effect [187].

The combination of HYP and mTHPC was also applied for the treatment of planktonic cultures of *Streptococcus mutans* and *Streptococcus sobrinus* [188]. These Gram-positive pathogens and their biofilms are among the main causes of dental caries [189,190] and are known for their resistance, which makes them difficult to completely eradicate, regardless of the method used, including single-PS PACT [188]. Studies have been conducted in vitro with mTHPC in liposomal formulation in concentrations from 0.625 to 10 μg/mL, and both single-PS PACT and combinations of two PSs were investigated [188]. For the photoactivation of the PSs, the same light sources and wavelengths (400–505 nm) were applied as those already used in dentistry for the polymerisation of dental materials, which are suitable for superficial applications [188]. Unexpectedly, the results were quite different between the two bacteria. One hundred percent of *S. sobrinus* was killed with HYP (already with 2.5 μg/mL) and mTHPC (5 μg/mL), as well as with the combination of the two PSs (1.25 μg/mL each) after 15 min of incubation and irradiation for 120 s (light density 1070 mW/cm^2^) [188]. On the other hand, there was no PACT effect against *S. mutans* with HYP, while already 1.25 μg/mL of mTHPC alone in PACT led to the 100% of the bacteria being killed; however, dark toxicity was observed from mTHPC for all investigated concentrations. The combination of PACT with HYP was only 99.9% effective at the highest concentration (10 μg/mL) and doubled incubation time (30 min) and irradiation (twice for 120 s); the same effectiveness was obtained with the two PSs in combination under conditions of 15 min of incubation (with 0.625 μg/mL for each PS) and 120 s irradiation [188]. It is known that HYP interacts with many biomolecules and shows many different bioactivities [191]. Therefore, the authors associated these results with the complex bioactivity of HYP and various mechanisms of its action in dark and light conditions. They also emphasised the advantage of using a combination of two PSs because, in such cases, the same PACT effect could be achieved in lower concentrations, thus avoiding dark toxicity [188].

A near-infrared dye, ICG [192], was combined with curcumin (Cur), a PS especially promising for PACT applications [193], to investigate another possible application in dentistry, i.e., their PACT effect against the biofilm of facultative bacterium *Enterococcus faecalis*, which is a cause of endodontic infections, to treat root canals more efficiently [194]. Curcumin is not water-soluble and is therefore often associated with low bioavailability, while ICG is unstable in water. Therefore, Cur was prepared as nanoparticles 200 nm in size for doping ICG, and metformin, otherwise known as a drug for type 2 diabetes, was also conjugated to increase their photosensitivity [194]. Two different light sources were used, a diode laser for irradiation at 810 nm (density 200 mW/cm^2^) for ICG and an LED-based source at 450 nm (500 mW/cm^2^) for Cur. Activity against *E. faecalis* was studied for each PS alone, with or without light, for the combination of the two PSs and for the combination of the two PSs and metformin with one light or two light sources and in two different orders of irradiation [194]. A significantly higher PACT effect (compared to single-PS treatments) achieved in the combination of all three drugs and both light sources was proven to be synergistic by post hoc Bonferroni test. The combination of all three drugs with one light source also resulted in a strong effect, which was stronger with an LED source than with a laser. In the case of using both light sources, the effect was slightly stronger when an LED source was applied first [194]. 

As in anticancer PDT, a synergistic effect in PACT is usually sought by using PSs that have different targets and mechanisms of action. Methylene blue (MB), a dye with a maximum absorption at 660 nm, is known as an active PS against both Gram-positive and Gram-negative bacteria [195] and may form an association with proteins and undergo the Type I process in PDT. Meanwhile, 6-carboxypterin (Cap) has a maximum absorption at 350 nm, undergoes both Type I and Type II processes in PDT and photodegrades to pterin [196]. Both were combined to treat a multidrug-resistant (MDR) strain of *Klebsiella pneumoniae*, an opportunistic Gram-negative bacterium resistant to multiple antibiotics [197]. *Klebsiella pneumoniae* was treated, both in planktonic form and in biofilm, with Cap (100 μM) and MB (2.5–10 μM) and irradiation (after 15 min of incubation) for 80 min with UV-A (365 nm, not exceeding a safe dose of 14.88 J/cm^2^) and visible light (350–750 nm) [196]. Small PDT and bacteriostatic effects were observed with the single PSs, while a synergistic effect was achieved in their combination. On the biofilm, the bactericidal effect continued to an even stronger extent than immediately after PDT, so 20 h after PDT, photokilling of almost 100% was reached with 100 μM Cap and 10 μM MB, and there was evidence that both Types I and II processes were involved [196].

To primarily utilise the Type II process, an MDR strain of *S. aureus* was treated with acetophenone-substituted phthalocyanines (Pcs) conjugated (via self-assembly through π-π interactions) to graphene quantum dots (GQDs), which are known as good ^1^O_2_ producers [198]. For this research, Pcs, which are dyes with strong absorption in the red and NIR regions, were prepared without metal or were metalated with zinc or indium; the highest ^1^O_2_ yield was measured for InPc (0.75), which was even higher for its conjugate with GQD (InPc-GQD) [198]. The activation of InPc-GQD containing 10 μM of InPc by irradiation at 670 nm led to 9.68 log reduction in the bacteria, while other conjugates had lower PACT effect than unconjugated Pc (with Zn or without metal), possibly due to aggregation, and the lowest activity was reported for non-metalated Pc conjugates [198].

Photodynamic action against microbes can also be used for disinfection, so photodynamic inactivation (PDI) is increasingly being researched in food decontamination as a new, low-cost and promising technology [199,200]. The natural pigments hypocrellin B (HB), which has a maximum absorption at 460 nm and may produce ROS through both Type I and Type II processes [201], and Cur, which has a maximum absorption at 420 nm and produces ROS through Type I process, were evaluated together in PACT on apples contaminated with the foodborne pathogen *S. aureus* [202]. Various concentrations of HB (from 0 to 500 nM), which did not cause dark toxicity, were tested with various light doses for photoactivation at 460 nm (from 0 to 9 J/cm^2^). An HB concentration of 500 nM and a light dose of 9 J/cm^2^ had the strongest bactericidal effect that led to increased membrane permeability and, consequently, the leakage of K^+^ and nucleic acids [202]. When sublethal Cur was used with HB, and 100 nM of each PS was irradiated to reach a light dose of 9 J/cm^2^ at both 420 nm (with light density 41.2 mW/cm^2^) and 460 nm (50.8 mW/cm^2^), a synergistic effect was observed for a dose of 1 J/cm^2^ only at 420 nm, but not at 460 nm, indicating the different cellular responses to HB and Cur and different types of ROS. However, although both Type I and Type II processes were involved, there was a higher production of ROS from Type I process [202]. The authors concluded that the apples were successfully decontaminated using PACT with HB and Cur against *S. aureus*, without affecting food quality as the apples retained the same phenolic content and pH [202].

A combination of two PSs, one of which is endogenously generated, has been used in several reported studies on *Leishmania*, a parasite that spreads by sandflies and causes a disease called leishmaniasis [203]. Leishmaniasis, most commonly cutaneous or mucocutaneous, is usually treated by chemotherapy, which is associated with side-effects and the development of resistance [204]. In one PDT study, uroporphyric mutants (DT) of *L. amazonensis* were modified to express relevant enzymes and produce uroporphyrin I (Uro I) induced by ALA and treated with ALA and aluminium phthalocyanine (AlPc) [205]. In vitro and mice studies were carried out with 1 mM of ALA and 0.01–1 ug/mL of AlPc, activated sequentially first by UV light (at 366 nm for the activation of Uro I) for 20 min, then with red light (above 650 nm for the activation of AlPc) for 5 min (light density 2.5 mW/cm^2^ and final dose 0.75 J/cm^2^), or only by white light for 1 h (10 J/cm^2^) [205]. The results indicated a synergistic effect explained by differences in the two PSs in their PDT mechanisms, subcellular localisations and different types of ROS produced, both in vitro and in vivo; however, the synergistic effect in vivo was only observed with white light and only in the ear dermis, probably due to limited light penetration [205]. In a more recent paper, drug-resistant promastigotes of *L. braziliensis* were also cloned to be responsive to ALA and produce Uro I, and compared to the wild-type (WT) *L. braziliensis*, which is not responsive to ALA. The observed viability was the lowest for the ALA-responsive *L. braziliensis* treated with ALA and UV light, and the highest viability was for the WT *L. braziliensis* [204]. In the treatment with two PSs, the ALA-responsive *L. braziliensis* was treated with ALA (1 mM), which was incubated for 48 h, and with diamino-phthalocyanine (diamino-Pc) (1 μM), incubated for 18 h, then irradiated with UV light for 30 min (to activate Uro I), followed by red light (~600 nm) for 60 min (for diamino-Pc) and, finally, macrophages loaded with the treated *Leishmania* [204]. The second strategy was different only in the way that the irradiation was applied after loading the macrophages. While both approaches resulted in a synergistic effect on the viability of *L. braziliensis* in macrophages, the procedure with irradiation after internalisation in macrophages had a slightly higher PDT effect and the highest increase in nitric oxide (NO) levels [204]. Furthermore, increased levels of CD40 and CD86 after PDT, reduced IL-10 and upregulated TNF and IL-6 supported the authors’ advocacy for the use of PDT for vaccination against leishmaniasis and their conclusion that, even though the second strategy was more successful, the first strategy would be safer and can be improved by increasing the doses [204].

Table 2 summarises the combinations of two PSs in antimicrobial PDT.

**Table 2 pharmaceuticals-16-00613-t002:** Summary of the combinations of two PSs in PACT/PDI with representative examples.

Photosensitisers	Microorganism	Irradiation Wavelength (Fluence Rate; Dose of Light)	Synergistic Effect	Other Observations/Outcomes	Ref.
HpD + mTHPC	*S. aureus* wild type	White light	Additive effect	Dark toxicity (mTHPC)	[186]
Photofrin + mTHPC	*S. aureus*	White light (60 mW/cm^2^; 100 J/cm^2^)	No	Antagonistic effect when HYP added	[187]
HYP + mTHPC	*S. mutans*; *S. sobrinus*	400–505 nm (1070 mW/cm^2^)	No	Reduced dark toxicity; HYP alone effective only against *S. sobrinus*	[188]
ICG + Cur (as NPs)	*E. faecalis* biofilm	Sequential: (1) 450 nm (Cur) (500 mW/cm^2^), (2) 810 nm (ICG) (200 mW/cm^2^)	Yes	Photosensitivity of both PSs increased by metformin	[194]
MB + Cap	MDR strain *K. pneumoniae* biofilm	365 nm (Cap) + 350–750 nm (MB)	Yes/100% bacterial photokilling	Both Type I and Type II PDT mechanism	[196]
HB + Cur	*S. aureus* on apples	420 nm (41.2 mW/cm^2^; 1 J/cm^2^)	Yes	Type I > Type II PDT	[202]
ALA (Uro I) + AlPc	*L. amazonensis* (in vitro and in vivo)	Sequential: (1) 366 nm (Uro I) (500 mW/cm^2^), (2) >650 nm (AlPc) (2.5 mW/cm^2^; 0.75 J/cm^2^) or only white light (10 J/cm^2^)	Yes (in vivo only with white light)	Both Type I and Type II PDT mechanism	[205]

## 5. Conclusions

In more than a century since its discovery, PDT has evolved mainly as an anticancer therapy; however, as an antimicrobial approach, it has been developing rapidly in recent years and shows considerable potential for various applications. The limitations of PDT, such as oxygen dependence and light penetration through tissue, have encouraged its use in combination with other therapies to take advantage of its benefits such as local activation and selectivity, different possible mechanisms of action and multiple targets. A specific combination therapy with PDT is the one that involves two or, rarely, even three different photosensitisers, which is the topic of this review.

The joint action of two photosensitisers in combination has so far been the most researched in anticancer PDT, in which the main approaches are based on evoking diverse types of cell death by targeting different organelles at the same time, such as the mitochondria, lysosomes and ER, as well as including both cellular and vascular targeting, with the aim of promoting adaptive immunity. Like Photofrin, BPD-MA most often plays a role in vascular targeting, but it is also used for targeting mitochondria. Endogenously produced PPIX from ALA, TPPS4, EtNBS and PDZ is mainly used for direct killing of cell (cellular targeting), and NPe6 is mainly used for targeting lysosomes. In sequential photoactivations, inflicting lysosomal photodamage before mitochondrial photodamage was shown to be more effective than in the inverted order due to the increase in pro-apoptotic signalling. Enhanced/synergistic PDT effects have been demonstrated both in vitro and in vivo, in some cases leading to a complete cure, even in the most aggressive types of cancer. In almost all cases, dark toxicity and skin photosensitivity were significantly reduced. Several reports showed that a synergistic effect in PDT was not linked to higher production of ^1^O_2_/ROS, suggesting the importance and potential of combining different cell death mechanisms.

Interest in using upconversion nanoparticles for anticancer PDT has particularly increased in the last ten years due to the possibility of using deep-penetrating red light, thus improving treatments in deep tissue. The combination of two PSs in UCNPs may overcome common deficiencies in UCNPs for PDT, such as low PS loading and low ROS production. In the UCNP studies with two PSs, the most frequently employed PSs so far were Ce6, which was used for the absorption of red emissions, and RB, which was used for the absorption of the green part of the upconversion emissions. 

Antimicrobial PDT with two PSs was investigated far less, albeit not very systematically, showing promising results, mostly on Gram-positive and Gram-negative bacteria, including resistant strains and their biofilms. A synergistic effect was shown on *Leishmania*, a protozoan parasite, after PACT with two PSs in a sequential protocol. The most usual strategy in all the described PACT studies was a combination of Type I and Type II processes to generate several types of ROS. Hypericin has been confirmed as a highly unpredictable PS with a complex mechanism of action in both anticancer and antimicrobial PDT, but has proven promising in combinations with hydrophobic PSs, as it appears to increase their photostability.

Despite the considerable number of benefits and encouraging results, research into these combinations is obviously very complex and this is probably the main reason development in this field proceeds so slowly and sporadically. It is quite demanding to develop standardised protocols and optimal PDT dosimetry with one PS, and clearly far more difficult to compare studies and results obtained with procedures that include two different irradiation wavelengths, light doses, DLI intervals and administrations of drugs and their photoactivation that can be simultaneous or occur in different sequences. These are the main challenges for future research worth taking on.

## Figures and Tables

**Figure 1 pharmaceuticals-16-00613-f001:**
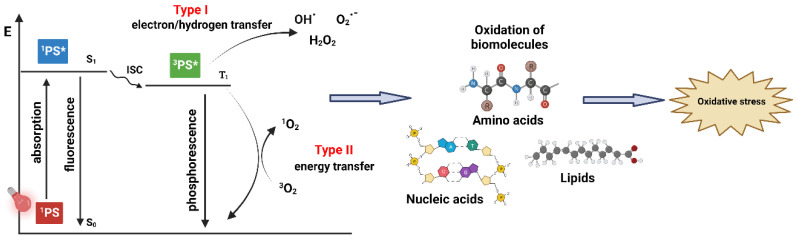
Generation of ROS by two types of processes in PDT leading to oxidative stress. Created with BioRender.com.

**Figure 2 pharmaceuticals-16-00613-f002:**
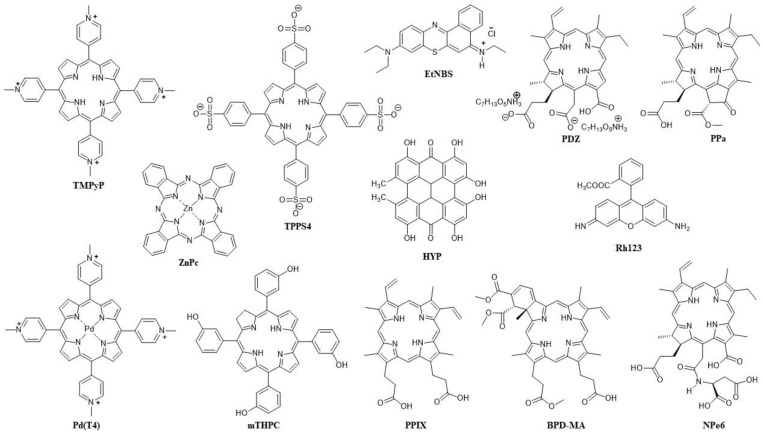
Structures of the PSs used in combination in anticancer PDT and their abbreviations.

**Figure 3 pharmaceuticals-16-00613-f003:**
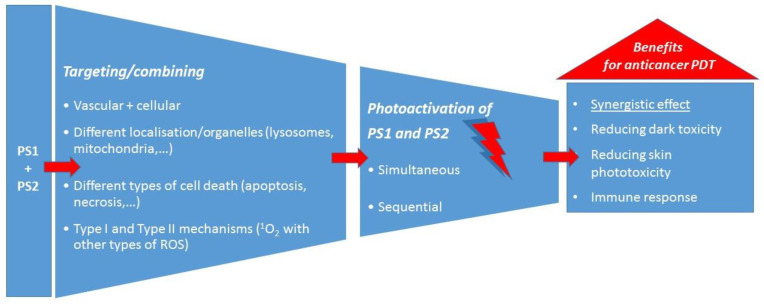
The main approaches in the use of two PSs for anticancer PDT and the main benefits.

**Figure 4 pharmaceuticals-16-00613-f004:**
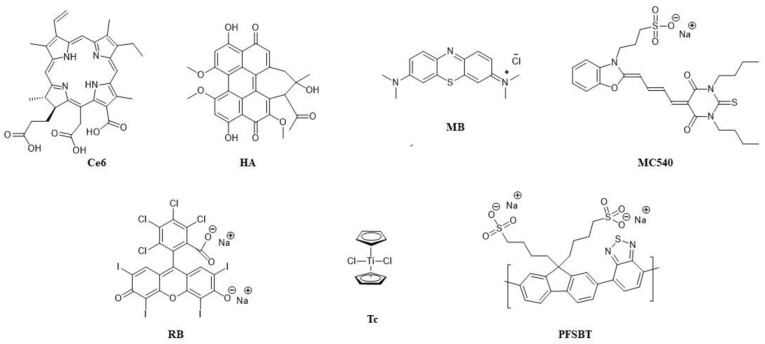
Structures of the PSs used for combining in UCNPs and their abbreviations.

**Figure 5 pharmaceuticals-16-00613-f005:**
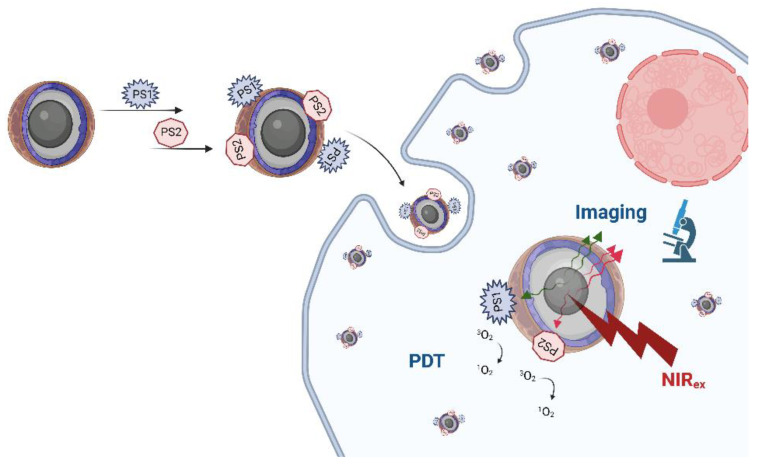
Illustration of the combination of two PSs in UCNPs for imaging and anticancer PDT. Created with BioRender.com.

**Figure 6 pharmaceuticals-16-00613-f006:**
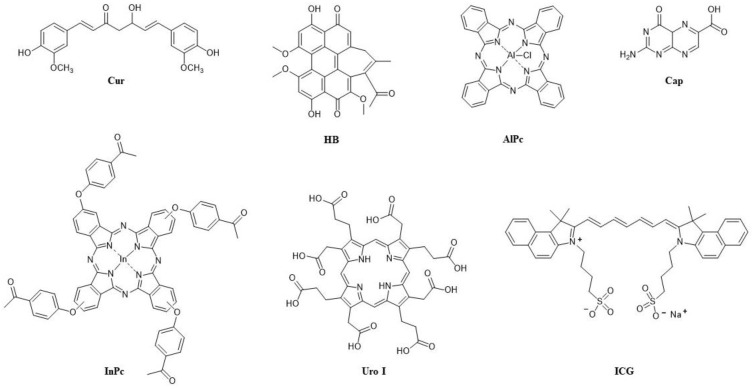
Structures of the PSs used in combination in PACT/PDI and their abbreviations.

## Data Availability

Not applicable.

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
