# Peer review of "Combination of Two Photosensitisers in Anticancer, Antimicrobial and Upconversion Photodynamic Therapy"

_pharmaceuticals, 2023, doi:10.3390/ph16040613_

Round 1

Reviewer 1 Report

The manuscript “Combination of Two Photosensitisers in Anticancer, Antimicrobial and Upconversion Photodynamic Therapy”  pharmaceuticals-2329667, is a very interesting piece of work which summarizes a large boy of information on the state of the art in the field of phototherapy, in particular in combined phototherapy especially in cancer treatment but also as antimicrobial treatment. It can be useful to readers from different areas of research both medical, biological, or chemical therefore deserves publication. Despite that, the conclusion section can be improved, it is not representative of the hole manuscript, and a few minor details must be fixed, see file attached.

Reviewer 2 Report

In this review paper the authors have discussed the importance of using two photosensitizers in PDT to overcome the shortcomings of the monotherapeutic approach. In addition, they have discussed the importance of combination therapy such as Upconversion Nanoparticles with Two PSs, Combining Two PSs in Photodynamic Antimicrobial Chemotherapy, and Combining Two PSs in PDT Against Cancer. The paper's subject is good, and the authors have tried to discuss several important issues. The manuscript is well-organized, and the topic area falls within the scope of Pharmaceutics, which can be considered for publication after minor revisions.

Comments: Major revision 

1.     The authors discussed several research findings. However, it is very important to include the figures from the experimental data to make it easily understandable.  Please add the most important result-finding figures under each section.

2.     The authors must add the nanocarrier section in the manuscript and discuss in detail the application of different nanocarrier in terms of multidrug delivery.

3.     The paragraph on page 11, lines 498-521, is about liposomal drug delivery.  Although two PSs are used in this experiment, they used liposome as the carrier and hence it would be better to move to the nanocarrier section.

Author Response

We are thankful to the Reviewer for all the comments and suggestions.

1) The authors discussed several research findings. However, it is very important to include the figures from the experimental data to make it easily understandable.  Please add the most important result-finding figures under each section.

RE: We agree completely with the Reviewer that a large number of different experimental data throughout the manuscript makes the text in some parts somewhat difficult to follow. There is Figure 3, which schematically summarizes the main approaches in the use of two photosensitisers for anticancer PDT and the main advantages (for section 2), and Figure 5 to illustrate the combination of two photosensitisers in upconversion nanoparticles for imaging and anticancer PDT (for section 3). Given the large number of different experimental data and outcomes, especially for anticancer PDT applications of combinations with two photosensitisers, we find it difficult to present them together with a single figure. Therefore, we added Table 1 and Table 2, to summarize the combinations of two photosensitisers in anticancer PDT (for section 2 without section 2.1) and PACT/PDI (for section 4) respectively, by comparing experimental data and results between representative examples. Also, in examples where several different conditions have been tested, only the most successful ones are listed in the table. We hope that each of these tables will provide an appropriate overview of the section, comparison and easier reading of the entire text.

2) The authors must add the nanocarrier section in the manuscript and discuss in detail the application of different nanocarrier in terms of multidrug delivery.

RE: We have devoted a significant part of the review to the description of nanocarriers in terms of multidrug delivery, but since it is part of a large section, we have now singled it out in a separate subsection (1.5.1. Nanocarriers for Combinations of PDT with Other Therapies) and reorganized it in order to get better focus.

3) The paragraph on page 11, lines 498-521, is about liposomal drug delivery.  Although two PSs are used in this experiment, they used liposome as the carrier and hence it would be better to move to the nanocarrier section.

RE: The Reviewer made a good point indicating the paragraph where one of the two photosensitisers (ZnPc) is delivered in liposome. We feel it is important to keep it in that section within the discussion on targeting different organelles, but we have added briefly mentioning it again in section 2.1. under ‘Conjugates and Nanocarriers for PSs’ Delivery and Imaging’. We would also like to point out that section 2.1. describes only examples where both photosensitizers are part of a conjugate or both delivered by nanocarriers.

Reviewer 3 Report

Dear authors, I have read the paper with very enthusiastically as this field of Photodynamic therapy (PDT) is very much emerging in biomedical treatments. From my point view this review paper is very important to understand the importance of PDT over those parameters. I believe that this review paper will attract the readership of similar field as well as the other readers who are all working in the field of biomedical treatments. Therefore, I recommend the acceptance of this review paper.

Author Response

Thank you very much for your comments.

Round 2

Reviewer 2 Report

The authors are addressed many of my previous questions and the manuscript is well improved. Hence, I recommend accepting for publication in the current forms.